# VisR-Bench: An Empirical Study on Visual Retrieval-Augmented Generation for Multilingual Long Document Understanding

## Abstract

Most organizational data in this world are stored as documents, and visual retrieval plays a crucial role in unlocking the collective intelligence from all these documents. However, existing benchmarks focus on English-only document retrieval or only consider multilingual question-answering on a single-page image. To bridge this gap, we introduce VisR-Bench, a multilingual benchmark designed for question-driven multimodal retrieval in long documents. Our benchmark comprises over 35K high-quality QA pairs across 1.2K documents, enabling fine-grained evaluation of multimodal retrieval. VisR-Bench spans sixteen languages with three question types (figures, text, and tables), offering diverse linguistic and question coverage. Unlike prior datasets, we include queries without explicit answers, preventing models from relying on superficial keyword matching. We evaluate various retrieval models, including text-based methods, multimodal encoders, and MLLMs, providing insights into their strengths and limitations. Our results show that while MLLMs significantly outperform text-based and multimodal encoder models, they still struggle with structured tables and low-resource languages, highlighting key challenges in multilingual visual retrieval.

## 1 Introduction

Retrieval-Augmented Generation (RAG) systems powered by Multimodal Large Language Models (MLLMs) Abootorabi et al. (2025); Zhu et al. (2024); Zhang et al. (2024a); Ghosh et al. (2024) have recently gained widespread attention in vision-and-language. In particular, retrieving accurate and relevant information from long documents poses unique challenges, as the systems should understand diverse structured content (e.g., tables, catalogs, figures) and capture complex document layouts Mathew et al. (2021c); Tanaka et al. (2023); Wu et al. (2024); Ding et al. (2024); Li et al. (2025a;b). Several MLLM-based retrieval models Jiang et al. (2024); Zhang et al. (2024b); Faysse et al. (2024); Chen et al. (2024b) have been introduced to address the challenges, yet their effectiveness remains largely untested. Existing retrieval benchmarks Mathew et al. (2022); Tito et al. (2023); Ma et al. (2024b); Wang et al. (2024); Dong et al. (2025) fall short in evaluation since they rely too heavily on text-image similarity rather than on Question-Answer (QA) relevance. An effective retrieval benchmark should require models to locate the information in the document that is relevant to the query and the answer, not

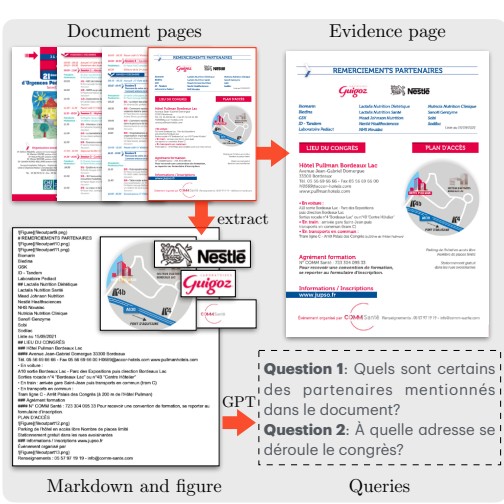

Figure 1: VisR-Bench is built by converting each PDF page into Markdown text and images. Queries are generated from this content, and during testing a retrieval model locates relevant pages, which a QA model then uses to answer the queries.

| Benchmark | Multi-page | Multiligual | QA | Retrieval | Text | Table | Figure | Doc # | Question # |
|---|---|---|---|---|---|---|---|---|---|
| M-LongDoc Chia et al. (2024) | ✓ | ✗ | ✓ | ✓ | ✓ | ✓ | ✓ | 300 | 10,070 |
| SlideVQA Tanaka et al. (2023) | ✓ | ✗ | ✓ | ✓ | ✓ | ✓ | ✓ | 2,619 | 14,484 |
| CVQA Romero et al. (2024) | ✗ | ✓ | ✓ | ✗ | ✗ | ✗ | ✓ | 5,239 | 10,374 |
| SciMMIR Wu et al. (2024) | ✓ | ✗ | ✗ | ✓ | ✓ | ✗ | ✗ | 530,000 | 530,000 |
| MMVQA Ding et al. (2024) | ✓ | ✗ | ✓ | ✓ | ✓ | ✗ | ✗ | 3,146 | 262,928 |
| DocMatix Laurençon et al. (2024) | ✗ | ✗ | ✓ | ✗ | ✓ | ✗ | ✗ | 2,444,750 | 9,500,000 |
| MMLongBench-Doc Ma et al. (2024b) | ✓ | ✗ | ✓ | ✓ | ✓ | ✓ | ✓ | 135 | 1,082 |
| MMDocIR Dong et al. (2025) | ✓ | ✗ | ✓ | ✓ | ✓ | ✓ | ✓ | 313 | 1,658 |
| DocVQA Mathew et al. (2021b) | ✗ | ✗ | ✓ | ✗ | ✓ | ✓ | ✓ | 12,767 | 50,000 |
| ChartVQA Masry et al. (2022) | ✗ | ✗ | ✓ | ✗ | ✓ | ✗ | ✓ | 21,945 | 32,719 |
| InfographicVQA Mathew et al. (2021a) | ✗ | ✗ | ✓ | ✗ | ✓ | ✓ | ✓ | 5,400 | 30,000 |
| ViDoRe Faysse et al. (2024) | ✓ | ✗ | ✓ | ✓ | ✓ | ✓ | ✓ | 8,310 | 3,810 |
| **VisR-Bench** | ✓ | ✓ | ✓ | ✓ | ✓ | ✓ | ✓ | 1,286 | 35,571 |

Table 1: Comparison with existing benchmarks. VisR-Bench is the first work on multi-page multilingual documents.

simply retrieving the visual content with the highest similarity to the query. For example, given the query, "When does the first train leave in the morning?", the relevant information should be retrieved from a train schedule table, not from an image of a train, despite the latter having the highest visual similarity to the query. In other words, the ideal retrieval benchmarks should go beyond surface-level similarities and incorporate deeper semantic and layout understanding. Besides, most prior Visual Question Answering (VQA) datasets Mathew et al. (2021c); Liu et al. (2024); Mathew et al. (2022); Van Landeghem et al. (2023) focus on QA tasks, assuming that the correct evidence image is provided. However, real-world retrieval scenarios are often more complex, not providing a single evidence page but instead documents with hundreds of images.

Another key challenge lies in the multilingual setting, especially in multimodal scenarios. For instance, most existing multilingual benchmarks focus on text-only document retrieval Tanaka et al. (2023); Wu et al. (2024); Ding et al. (2024); Ma et al. (2024a), offering limited insights into "multimodal" retrieval performance. Likewise, most prior multimodal retrieval benchmarks rely on the English-only setting Ma et al. (2024b); Dong et al. (2025); Chia et al. (2024) without considering "other" languages. These limitations further hinder a comprehensive evaluation of MLLMs' retrieval capabilities.

To address these challenges, we propose **VisR-Bench**, the first question-driven multilingual **Vis**ual **R**etrieval benchmark designed to evaluate MLLM retrieval performance in visually rich document images. VisR-Bench consists of 53K high-quality synthetic QA pairs and 1,286 documents (373 English and 913 multilingual), with an average length of about eighteen pages. By generating QA pairs for different evidence types, tables, figures, and visual text, our benchmark enables granular performance analysis in multimodal reasoning, OCR robustness, and table understanding. As illustrated in Figure 2, we collect multimodal documents from sixteen languages, such as English and Italian, allowing for the assessment of language-specific weaknesses in existing retrievers and an extensive evaluation of multilingual retrieval abilities. Additionally, English documents are categorized into ten document types (e.g., newsletter, magazine).

We summarize our key contributions as follows:

- We introduce **VisR-Bench**, a benchmark that systematically evaluates the retrieval capabilities of MLLMs in multilingual and multimodal settings. Our dataset spans sixteen languages and encompasses diverse document types.

- We evaluate a wide range of retrieval models, including text-based methods, multimodal encoders, and large MLLMs, providing quantitative insights into retrieval performance across different evidence types and languages.

- We show that MLLMs significantly outperform text-based and vision-language encoder models but still struggle with structured documents and low-resource languages, revealing language-specific and layout-specific challenges and providing insights for improving MLLMs.

## 2 RELATED WORK

**Text-based Retrieval Methods** Traditional text-based retrieval methods extract text from images using OCR tools (Du et al., 2020; Singh et al., 2021) and apply text-based retrieval techniques. BM25

(Robertson et al., 2009) is a statistical algorithm based on text frequency. Deep learning models such as Sentence-BERT (Reimers, 2019) and BGE Models (Chen et al., 2024c; Xiao et al., 2023) enable semantically aware search. NV-Embed (Lee et al., 2024), built upon LLMs (*e.g.*, Mistral 7B (Jiang et al., 2023)), generates text embedding to enhance retrieval accuracy by capturing richer contextual information. However, these approaches struggle with complex layouts and cannot process visual elements, limiting their performance in real-world applications.

**Multimodal Retrieval Methods**    Multi-modal encoders like CLIP (Radford et al., 2021) and SigLIP (Zhai et al., 2023) can be used for image retrieval by similarity in a shared embedding space, but they are optimized for natural images rather than document pages. With the advent of MLLMs, recent approaches customize MLLMs as encoders, leveraging their pre-trained knowledge for improved accuracy. VLM2Vec (Jiang et al., 2024) and GME (Zhang et al., 2024b) compute similarity using single-vector embeddings, while ColPali (Faysse et al., 2024), ColPhi (Chen et al., 2024b), and ColInternVL2 (Chen et al., 2024b) utilize sequences of hidden states and apply sequence interaction scoring (Khattab & Zaharia, 2020) for more effective relevance estimation.

**Comparison with Multi-page Datasets**    Existing multi-page document datasets focus on domain-specific documents, such as SlideVQA (Tanaka et al., 2023), SciMMIR (Wu et al., 2024), and MMVQA (Ding et al., 2024). Wiki-SS (Ma et al., 2024a) emphasizes text-based evidence. DocMatix (Laurençon et al., 2024) contains noisy and ambiguous queries, and CVQA (Romero et al., 2024) is limited to single natural images, making it unsuitable for document retrieval. Additionally, MMLongBench-Doc (Ma et al., 2024b), MMDocIR (Dong et al., 2025), and M-LongDoc (Chia et al., 2024) are English-only, limiting the multilingual applicability.

## 3    VISR-BENCH

The VisR-Bench is divided into an English Multimodel Split (English only) curated from web-crawled data and a Multilingual Multimodel Split (15 non-English languages) filtered from the CCpdf dataset (Turski et al., 2023). As demonstrated in Table 1, VisR-Bench is a comprehensive benchmark that integrates multiple essential features of multilingual visual retrieval tasks. Unlike existing benchmarks, VisR-Bench simultaneously supports multilingual documents while being suitable for evaluating retrieval functionalities across diverse content types, including text, tables, and figures. Its large scale, spanning 16 languages, 1,286 documents, and 35,571 questions, makes it a robust resource for developing and evaluating models capable of handling complex, real-world document analysis challenges.

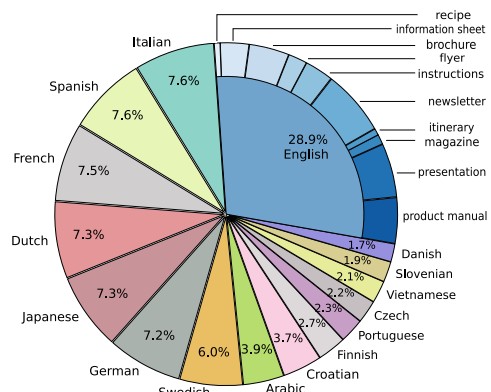

Figure 2: Distribution of language and document types in VisR-Bench. The blue colors represent the English multimodal split, which can further be categorized into ten document types. Other colors represent the multilingual multimodal split, containing documents in fifteen non-English languages.

### 3.1    DATA SOURCING

We start from a large-scale, diverse, multilingual corpus of PDF files from all over the Internet using Common Crawl (Turski et al., 2023). All documents are extracted using a document parser[1]. We excluded documents with PDF or quality issues and obtained 301,553 documents. The document parser outputs markdown files, including texts, tables, and figures for each document page in an interleaved manner. All figures are saved separately as images with a path reference in markdown files. Document extraction examples are provided in Figure 3. We will release all these markdown files for future document research.

---

[1]Adobe Document-Extract-API: https://developer.adobe.com/document-services/apis/pdf-extract/

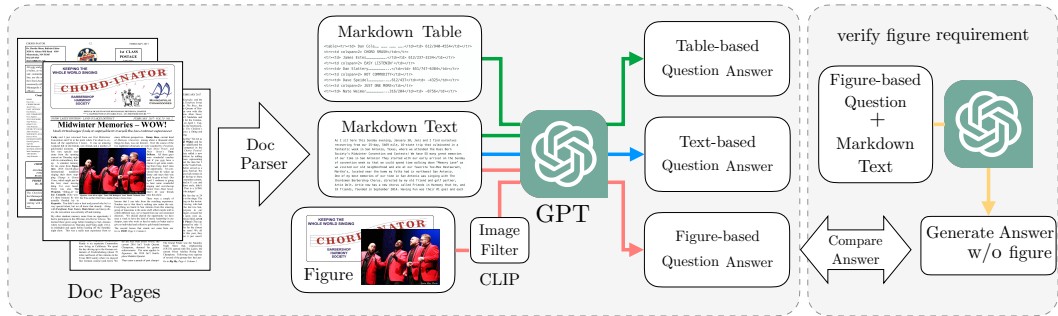

Figure 3: **Overview of our QA data generation pipeline.** Text and visual content are first extracted from documents, with regular text and tables saved as Markdown—tables are preserved in structured text format using Markdown table syntax—while figures are saved as separate image files. For table- and text-based QA, we prompt GPT-4o using the extracted Markdown content. For figure-based QA, we first filter out decorative figures using a CLIP-based classifier, then generate figure-centered questions by prompting GPT-4o with only the image. To ensure that figures are truly required for answering, we revise the QA pairs by incorporating surrounding text and apply a heuristic filtering step: any question that GPT-4o can already answer using only the Markdown text is discarded. This ensures that the final figure-based QA pairs require both visual and textual information for accurate retrieval and comprehension.

## 3.2 ENGLISH MULTIMODEL SPLIT

To construct the English multimodel split, we select about $4,000$ PDF documents from the filtered ccPDF documents, which are from 38 types of documents based on human annotations. To ensure a focus on multimodal content, we select ten types of documents that are visually rich, such as product manuals and presentations. Further filtering is applied to retain only English documents that contain both Markdown files and informative figures while excluding single-page documents, as retrieval is unnecessary for them. The final dataset includes 210 table-rich documents, 310 text-rich documents, 125 figure-rich documents, and 913 multilingual documents, ensuring a balanced evaluation across different retrieval types. Figure 2 and Figure 4 show the distribution of the types of human-annotated documents and their average lengths, highlighting a greater diversity than previous benchmarks (Tanaka et al., 2023; Islam et al., 2023; Ma et al., 2024b).

To select documents with informative figures, we apply figure classification on the extracted images using the CLIP model `ViT-L/14-336` (Radford et al., 2021). Each figure is classified into one of 19 predefined categories, and we retain 6 relevant types while discarding decorative figures such as logos and banners. After filtering, the multimodal evaluation split is refined to 373 unique documents. All documents have been validated by human reviewers to ensure the exclusion of harmful content and personally identifiable information (PII). Furthermore, we confirm that the license and usage terms of each document explicitly allow its use for research purposes. To build a multimodal document retrieval benchmark, we categorize all questions into figure-related, text-related, and table-related questions, and use different prompting strategies using GPT-4o Hurst et al. (2024) to curate questions.

**Figure-related QA** We combine the figures with their corresponding contexts and use GPT-4o (API version 2024-08-15) to generate QA pairs. For prompt construction, we provide two demonstrations and instruct GPT-4o to generate a new QA pair. To ensure that the figures are necessary to answer the questions, we apply a heuristic filtering step: we discard any question that GPT-4o can already answer using only the textual information extracted from the Markdown files, as shown in Figure 3. This process not only enforces reliance on visual content, but also serves as an additional validation step for the correctness of the generated answers. In contrast, although many existing benchmarks in Table 1 include figure-based questions, they do not isolate them or verify whether the figure is actually required to answer.

**Text-based QA** To generate text-based QA pairs, we first filter pages that contain only text in the extracted Markdown files, excluding those with tables or figures to ensure a sole focus on textual information. We then use GPT-4o to generate QA pairs over the given page. We design a system prompt to enforce key constraints: (1) Questions must simulate a realistic retrieval scenario where a

user queries a multi-page document for relevant information. (2) Answers must be explicitly present in the text to prevent hallucination. (3) Questions should not be ambiguous or overly broad, such as asking for the page number or requiring document-level summarization. (4) If a page lacks sufficient content for meaningful questions, the model returns an empty string instead of generating forced or unnatural queries.

**Table-related QA**    Similar to text-based QA, we extract pages that contain tables but no figures to ensure that the generated questions are not influenced by visual elements. This guarantees that the QA pairs focus solely on tabular data and its text context. In addition to the constraints applied to text-based QA, table-related questions are designed to require computation or logical inference rather than simple fact lookup. Instead of directly extracting a single value, the questions encourage tasks such as analyzing trends, making comparisons, identifying rankings, or interpreting correlations within the table data. This ensures that retrieval models must engage in structured reasoning.

### 3.3 MULTILINGUAL MULTIMODEL SPLIT

Our dataset includes multilingual queries over documents in 15 non-English languages, including Spanish, Italian, German, French, Dutch, Arabic, Croatian, Japanese, Swedish, Vietnamese, Portuguese, Finnish, Czech, Slovenian, and Danish, as shown in Figure 2. These fifteen languages are selected based on our filtered documents, with the number of associated documents more than 500. This subset is designed to assess the accuracy of the retriever in a diverse linguistic landscape. The queries are general questions generated by GPT-4o, conditioned on text, tables, and figures. To simplify human inspection of the generated QA pairs, we curated a prompt to generate questions in both English and another language. Detailed prompts are provided in the Appendix B.

**Multilingual Finetuning Data**    Previous retriever models Faysse et al. (2024); Chen et al. (2024b) were typically fine-tuned on data biased toward English, which may lead to reduced performance in other languages. To investigate whether multilingual training could improve performance, we scaled the data generation process described above to produce a larger multilingual dataset for fine-tuning, containing 210k QA pairs and 39.5k documents.

## 4 EXPERIMENTAL RESULTS

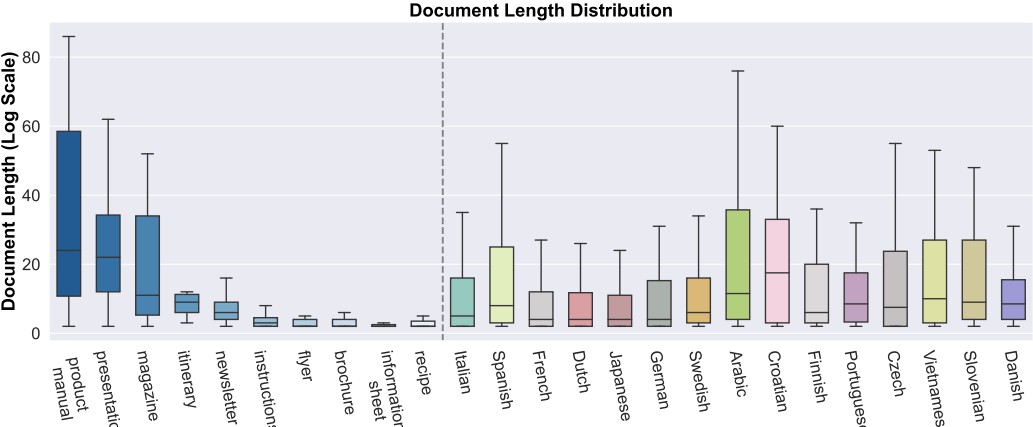

Figure 4: Boxplot of document length distribution. Each box represents the inter-quartile range (IQR), covering the middle $50\%$ of the data. The horizontal line inside each box indicates the median document length, while the whiskers extend to the minimum and maximum values within 1.5 times the IQR. The dashed vertical line separates English multimodal split (left) from multilingual multimodal documents (right).

### 4.1 EVALUATION SUITE

**Top-$k$ Retrieval Accuracy**    Since all QA samples in the VisR-Bench dataset require a single evidence page, we evaluate document retrieval using top-1 and top-5 Accuracy. This binary metric

assigns a score of 1 if the ground-truth evidence page appears in the first and top five retrieved results and 0 otherwise. The final accuracy is the percentage of samples with a score of 1, directly measuring retrieval effectiveness in this setting.

**PNLS**   We adopt PNLS Chen et al. (2024a) as a metric to evaluate the similarity between the model-generated answer and the ground truth. PNLS is a variant of normalized Levenshtein similarity Biten et al. (2019) that identifies an optimally aligned substring in the ground truth using dynamic programming. It then measures the edit distance between this substring and the model-generated answer, normalizing by the length of the aligned substring (including matches and gaps). This normalization ensures that concise responses are not unfairly penalized, making PNLS particularly suitable for evaluating long-form answers and cases where partial correctness matters.

**GPT Evaluation**   For long or complex answers, string-matching metrics fail to provide accurate evaluation. Instead, we use GPT-based evaluation, a binary metric where GPT compares the model's answer with the ground truth. If they convey the same information, the sample receives a score of 1, otherwise 0. The average score across samples is reported as GPT accuracy (GAcc), offering a more reliable assessment beyond exact string matching.

## 4.2   RETRIEVAL RESULTS ON ENGLISH SPLIT

The retrieval performance of 14 different methods on the English English split is shown in Table 2. Retrieval methods are categorized into (1) Text-based Methods, (2) Multimodel Encoders, and (3) Multimodel Large Language Models. These results highlight the key trends in multimodal document retrieval, revealing the strengths and limitations of different retrieval approaches across figures, tables, and text. Below are some findings from our experimental results.

| | | Figure | | Table | | Text | | Average | |
|---|---|---|---|---|---|---|---|---|---|
| | Accuracy | top1 | top5 | top1 | top5 | top1 | top5 | top1 | top5 |
| *Text-based Methods* | | | | | | | | | |
| BM25 (Chen et al., 2024c) | | 24.27 | 45.63 | 38.58 | 66.43 | 64.72 | 89.10 | 42.52 | 67.05 |
| SBERT (Reimers, 2019) | | 25.24 | 49.27 | 26.31 | 52.68 | 49.96 | 76.97 | 33.84 | 59.64 |
| BGE-large (Xiao et al., 2023) | | 31.55 | 56.07 | 40.36 | 70.14 | 57.00 | 82.68 | 42.97 | 69.63 |
| BGE-M3 (Chen et al., 2024c) | | 31.07 | 56.80 | 51.11 | 78.51 | 67.70 | 89.89 | 49.96 | 73.95 |
| NV-Embed-v2 (Lee et al., 2024) | | 35.44 | 65.05 | 44.04 | 73.34 | 61.38 | 87.46 | 46.95 | 75.28 |
| *Multimodal Encoders* | | | | | | | | | |
| CLIP (Radford et al., 2021) | | 33.90 | 61.74 | 24.68 | 47.59 | 39.47 | 70.21 | 32.68 | 59.85 |
| SigLip (Zhai et al., 2023) | | 38.98 | 69.73 | 24.73 | 53.22 | 39.06 | 70.97 | 34.26 | 64.64 |
| *Multimodal Large Language Models* | | | | | | | | | |
| VisRAG (Yu et al., 2024) | | 31.96 | 66.83 | 19.82 | 48.53 | 31.00 | 61.49 | 27.59 | 58.95 |
| VLM2Vec (Jiang et al., 2024) | | 40.44 | 76.27 | 28.51 | 57.77 | 39.90 | 71.69 | 36.28 | 68.58 |
| GME (Zhang et al., 2024b) | | 68.04 | 91.53 | 61.50 | 86.38 | 76.34 | 95.62 | 68.63 | 91.18 |
| Col-InternVL2 (Chen et al., 2024b) | | 68.28 | 90.31 | 63.85 | 86.36 | 79.19 | 96.45 | 70.44 | 91.04 |
| Col-Phi (Chen et al., 2024b) | | 68.77 | 93.22 | 65.65 | 88.51 | 81.67 | 97.04 | 72.03 | 92.92 |
| ColPali-v1.2 (Faysse et al., 2024) | | 68.77 | 91.77 | 66.12 | 88.26 | 82.63 | 96.89 | 72.51 | 92.31 |
| ColQwen2-v0.1 (Faysse et al., 2024) | | **74.58** | **95.64** | **67.43** | **88.98** | **83.68** | **97.61** | **75.23** | **94.08** |

Table 2: Retrieval accuracy results on VisR-Bench(English split). Bold font indicates the best overall performance for each language.

**Even the best method does not perform perfectly in VisR-Bench, indicating the difficulty of our benchmark and the substantial room for improvement.**   By examining the top-1 performances across different methods, we observe that even the best method, ColQwen2[2], can only reach 75.23% of average accuracy, while the performances are even worse on figure and table content retrieval. This phenomenon not only shows the difficulty of our benchmark but also indicates the large space for future models.

**Retrieval of table content is still challenging for multimodal encoders and MLLMs.**   Multimodal encoders and MLLMs-based methods consistently perform poorer on table content retrieval compared with text content and figure content, probably due to the different information processing process. This phenomenon indicates that structured tabular data still poses unique perceiving, understanding, and retrieval challenges that are not fully addressed by existing models. The results suggest that tables require specialized retrieval mechanisms beyond standard embeddings, emphasizing the need for better table-aware perceiving and retrieval techniques.

---

[2]huggingface: https://huggingface.co/vidore/colqwen2-v0.1

**MLLM-based methods outperform other methods consistently.** Our results show that MLLM-based retrieval methods consistently outperform all other methods with a large margin, demonstrating their advantage in end-to-end document understanding and retrieval. Specifically, ColQwen2 achieves the highest retrieval accuracies across figures, tables, and text. Despite identical data and protocols, it surpasses ColPali by $\sim 2\%$, suggesting that base-model pretraining quality plays a key role in this task. Among the remaining MLLMs, ColPali and ColPhi perform comparably, while ColInternVL2 and GME underperform slightly. VisRAG and VLM2Vec perform poorly, likely due to their optimization for natural images rather than document structures. Meanwhile, without surprise, text-based methods perform promisingly well on text retrieval but struggle with figures and tables, confirming the limitations of text-only approaches in multimodal retrieval.

**Contextualized late interaction outperforms single-vector similarity.** Although trained on massive image datasets, multimodal encoders such as CLIP and SigLip fall well behind MLLMs, suggesting that vision–language pretraining alone is insufficient and that deeper contextual reasoning, as enabled by contextualized late interaction, is crucial for effective multimodal retrieval. This distinction is further illustrated by the gap between multi-vector and single-vector embedding models: ColQwen2, a multi-vector model built on the smaller Qwen2-VL-2B, significantly outperforms GME, a single-vector model based on the larger Qwen2-VL-7B. These results underscore that capturing finer-grained, context-dependent representations through late interaction can outweigh even substantial differences in base model size.

**LLM Retrievers Excel at Figures and Tables.** As shown in the table, NV-Embed-v2—a recent 7B LLM-based retriever—demonstrates clear advantages over BM25 in figure- and table-based QA. Although it operates solely on Markdown input, its strong performance in these settings may be attributed to its language modeling capabilities and pretrained knowledge of figure- and table-related concepts. This enables it to infer implicit information even without direct visual input. These results highlight the potential of LLM-based retrievers to reason over semi-structured content by leveraging contextual cues and prior knowledge, particularly in scenarios where information is weakly grounded in text.

### 4.3 RETRIEVAL RESULTS ON MULTILINGUAL SPLIT

Table 3 shows the retrieval performance of 15 different methods, including a fine-tuned ColQwen2-v0.1 model on the multilingual training set described in section 3.3, evaluated across 15 non-English languages in the Multilingual English split. For clarity, we only present the average accuracy for each language without splitting them into different content sources. The performance on this multilingual split for the first time shows how different methods perform under this challenging multilingual multimodal retrieval scenario. Below are some findings from our experimental results.

**Most Methods Struggle on Low-Resource Languages.** The retrieval accuracy results show a clear gap in performance between different languages, particularly when comparing well-resourced languages like Spanish, Italian, and German to low-resource languages such as Arabic, Finnish, and Vietnamese. Across all model categories, text-based methods, multimodal encoders, and MLLMs, the accuracy scores drop significantly for low-resource languages. This indicates that despite the advances in current retrieval methods, language resource availability continues to play a critical role in performance, and models still struggle to generalize well to underrepresented languages. This phenomenon highlights the importance of our benchmark.

**MLLMs and Encoders still face multilingual challenges.** Despite recent progress, both MLLM-based retrieval methods (e.g., ColQwen) and multimodal encoders (e.g., CLIP and SigLIP) exhibit clear limitations in low-resource language settings. MLLMs perform well in some languages. ColQwen2 achieves the best accuracy in several cases—but are inconsistent overall, often being outperformed by text-based methods like BM25 and BGE-M3, especially in Czech and Portuguese. Meanwhile, CLIP and SigLIP consistently underperform across nearly all low-resource languages, with significantly lower top-1 accuracy compared to both MLLMs and text-only methods. These results suggest that neither current MLLMs nor multimodal encoders are robustly optimized for multilingual scenarios, highlighting the need for improved multilingual training and evaluation across both unimodal and multimodal retrieval systems.

**Text-based methods Beating MLLMs in Multilingual.** Although MLLMs have shown promising results, text-based retrieval methods, especially those tailored for multilingual settings, remain

| | Spanish | | Italian | | German | | French | | Dutch | | Arabic | | Croatian | | Japanese | |
|---|---|---|---|---|---|---|---|---|---|---|---|---|---|---|---|---|
| Accuracy | top1 | top5 | top1 | top5 | top1 | top5 | top1 | top5 | top1 | top5 | top1 | top5 | top1 | top5 | top1 | top5 |
| *Text-based Methods* | | | | | | | | | | | | | | | | |
| BM25 | 60.25 | 82.50 | 59.14 | 82.02 | 65.82 | 86.92 | 54.07 | 77.79 | 59.83 | 84.88 | 7.43 | 21.49 | 52.98 | 72.71 | 11.59 | 38.60 |
| SBERT | 22.77 | 41.83 | 21.82 | 41.12 | 25.74 | 48.54 | 27.43 | 51.33 | 27.99 | 52.25 | 4.02 | 17.29 | 17.72 | 36.67 | 13.06 | 41.24 |
| BGE-large | 34.55 | 60.41 | 30.27 | 56.24 | 39.75 | 66.82 | 41.34 | 67.42 | 39.14 | 67.53 | 6.15 | 19.53 | 32.67 | 58.14 | 31.92 | 64.97 |
| BGE-M3 | 58.16 | 83.13 | 52.94 | 77.96 | 67.64 | 88.94 | 60.68 | 82.10 | 63.62 | 87.73 | 10.55 | 26.26 | **59.07** | **81.46** | 58.38 | 84.33 |
| NV-Embed-v2 | 42.92 | 72.71 | 40.84 | 66.32 | 52.23 | 80.30 | 49.41 | 76.13 | 47.12 | 78.74 | 5.47 | 21.73 | 41.86 | 68.30 | 42.17 | 72.70 |
| *Multimodal Encoders* | | | | | | | | | | | | | | | | |
| CLIP | 11.14 | 29.32 | 12.39 | 31.77 | 19.53 | 45.69 | 19.52 | 44.44 | 16.22 | 42.71 | 4.64 | 18.91 | 10.46 | 27.36 | 14.28 | 44.86 |
| SigLIP | 13.08 | 32.36 | 17.52 | 40.69 | 25.69 | 51.69 | 24.85 | 53.15 | 22.70 | 50.85 | 5.53 | 19.56 | 13.98 | 33.56 | 15.62 | 46.20 |
| *Multimodal Large Language Models* | | | | | | | | | | | | | | | | |
| VisRAG | 9.70 | 28.48 | 10.69 | 33.09 | 14.48 | 40.22 | 16.37 | 42.55 | 15.22 | 42.02 | 4.78 | 19.80 | 6.38 | 22.25 | 21.04 | 52.37 |
| VLM2Vec | 18.59 | 44.48 | 19.42 | 43.84 | 26.07 | 56.10 | 29.53 | 60.50 | 22.51 | 52.97 | 7.39 | 24.10 | 12.31 | 32.04 | 19.19 | 50.02 |
| GME | 60.57 | 88.08 | 52.96 | 79.08 | 65.97 | 89.61 | 66.78 | 89.55 | 57.92 | 85.16 | **15.33** | **35.72** | 45.09 | 72.60 | 61.11 | 89.37 |
| ColInternVL2 | 58.26 | 84.57 | 51.89 | 77.96 | 60.35 | 86.32 | 64.06 | 87.17 | 58.27 | 84.60 | 5.09 | 17.50 | 47.68 | 73.16 | 39.65 | 71.57 |
| ColPhi | 65.42 | 89.00 | 56.06 | 81.43 | 65.02 | 88.96 | 67.83 | 89.65 | 62.15 | 88.17 | 8.46 | 25.95 | 48.83 | 74.82 | 25.28 | 56.49 |
| ColPali-v1.2 | 71.44 | 92.62 | 62.02 | 85.81 | 72.96 | 92.48 | 72.62 | 92.09 | 65.15 | 89.73 | 14.33 | 32.59 | 51.54 | 76.94 | 43.85 | 77.53 |
| ColQwen2-v0.1 | **75.04** | **94.34** | **65.18** | **88.24** | **78.63** | **95.77** | **77.81** | **93.69** | **70.30** | **92.12** | 12.05 | 27.16 | 55.27 | 79.20 | **65.81** | 89.41 |
| *Finetuned on Multilingual Data* | | | | | | | | | | | | | | | | |
| ColQwen2 (E) | 67.25 | 90.60 | 57.10 | 82.29 | 71.99 | 93.18 | 72.01 | 91.39 | 60.26 | 86.57 | 10.15 | 26.70 | 44.50 | 71.12 | 61.54 | 87.70 |
| ColQwen2 (M) | 69.77 | 92.48 | 59.77 | 85.39 | 72.71 | 92.97 | 72.70 | 92.16 | 64.37 | 89.10 | 11.67 | 28.07 | 50.44 | 76.83 | 63.91 | **89.98** |

| | Swedish | | Vietnamese | | Portuguese | | Finnish | | Czech | | Slovenian | | Danish | | Average | |
|---|---|---|---|---|---|---|---|---|---|---|---|---|---|---|---|---|
| *Text-based Methods* | | | | | | | | | | | | | | | | |
| BM25 | 57.44 | 83.68 | **48.81** | **73.01** | 61.47 | 79.92 | 50.11 | 71.24 | 66.11 | 89.34 | 56.45 | 81.81 | 54.38 | 82.35 | 52.38 | 74.82 |
| SBERT | 28.26 | 60.99 | 17.94 | 37.07 | 25.85 | 50.24 | 23.34 | 47.29 | 26.28 | 50.00 | 22.31 | 48.03 | 29.56 | 58.11 | 22.05 | 43.98 |
| BGE-large | 42.18 | 74.69 | 23.94 | 48.97 | 38.53 | 66.08 | 31.58 | 57.97 | 33.97 | 61.94 | 35.30 | 63.89 | 35.58 | 71.02 | 33.33 | 59.75 |
| BGE-M3 | 65.25 | 89.33 | 44.93 | 68.82 | 60.07 | 82.16 | **56.90** | **77.19** | **65.87** | 90.22 | **65.05** | 88.53 | 64.42 | 88.38 | 56.25 | 79.34 |
| NV-Embed-v2 | 53.02 | 81.40 | 25.75 | 60.24 | 56.98 | 80.34 | 34.32 | 61.63 | 41.99 | 70.59 | 43.91 | 73.21 | 52.94 | 79.48 | 42.03 | 69.63 |
| *Multimodal Encoders* | | | | | | | | | | | | | | | | |
| CLIP | 17.38 | 48.84 | 6.67 | 22.13 | 16.75 | 42.17 | 12.13 | 36.84 | 11.86 | 34.78 | 13.35 | 36.29 | 13.77 | 45.48 | 13.40 | 35.60 |
| SigLIP | 26.78 | 61.66 | 8.38 | 25.13 | 25.30 | 51.03 | 17.24 | 45.84 | 20.67 | 47.92 | 17.03 | 43.28 | 23.53 | 54.38 | 17.87 | 41.87 |
| *Multimodal Large Language Models* | | | | | | | | | | | | | | | | |
| VisRAG | 14.93 | 49.30 | 5.53 | 18.56 | 12.68 | 38.29 | 9.76 | 34.78 | 9.46 | 34.13 | 8.69 | 34.50 | 13.77 | 46.34 | 11.61 | 34.61 |
| VLM2Vec | 25.98 | 62.17 | 8.22 | 25.39 | 23.73 | 53.46 | 16.17 | 44.47 | 21.07 | 50.56 | 15.59 | 44.09 | 25.39 | 58.68 | 19.72 | 46.57 |
| GME | 59.09 | 89.79 | 26.22 | 51.81 | 65.29 | 90.72 | 38.83 | 68.80 | 51.52 | 82.29 | 51.34 | 80.91 | 54.52 | 86.94 | 53.85 | 80.26 |
| ColInternVL2 | 61.16 | 90.51 | 25.75 | 54.19 | 62.32 | 86.95 | 46.22 | 72.85 | 55.29 | 85.90 | 54.75 | 83.96 | 60.11 | 90.10 | 51.29 | 77.04 |
| ColPhi | 64.19 | **93.08** | 34.28 | 65.25 | 64.99 | 88.59 | 49.73 | 75.82 | 58.65 | 88.86 | 56.81 | 85.66 | 61.12 | 90.67 | 54.71 | 79.84 |
| ColPali-v1.2 | 65.37 | 92.11 | 35.32 | 66.60 | 76.03 | 92.96 | 42.11 | 73.07 | 62.34 | 91.27 | 55.82 | 86.29 | 62.41 | 90.10 | 60.00 | 83.65 |
| ColQwen2-v0.1 | **70.16** | **95.37** | 35.39 | 64.51 | **76.32** | **93.53** | 49.72 | 75.82 | 65.03 | **92.57** | 61.67 | 88.98 | **72.31** | **94.76** | **62.04** | **84.35** |
| *Finetuned on Multilingual Data* | | | | | | | | | | | | | | | | |
| ColQwen2 (E) | 62.30 | 91.37 | 24.50 | 53.29 | 49.44 | 80.19 | 40.88 | 68.43 | 52.87 | 84.12 | 49.44 | 80.19 | 63.92 | 92.22 | 53.84 | 79.32 |
| ColQwen2 (M) | 66.12 | 94.05 | 27.10 | 55.05 | 71.86 | 92.09 | 40.88 | 70.04 | 61.32 | 89.63 | 58.43 | **89.35** | 64.52 | 93.22 | 56.07 | 82.45 |

Table 3: Retrieval accuracy results on VisR-Bench(multilingual split). Bold font indicates the best overall performance for each language.

competitive and, in some cases, superior. BGE-M3, a multilingual text-based method, achieves the best performance in several languages, such as Finnish and Czech, surpassing all MLLM by a considerable margin. Similarly, BM25, a traditional text retrieval method, performs surprisingly well in languages like Vietnamese, outperforming many large models.

**Challenges of Arabic for Retrieval Models.** Arabic remains one of the most challenging languages, with retrieval accuracy far below that of others. Even top models like ColQwen2 and GME, as well as text-based methods such as BM25 and BGE-M3, perform poorly. This may stem from its rich morphology, complex script, distinct syntax, and right-to-left reading order, which may require dynamic designs in attention masks and position embeddings. Addressing these issues could involve language-specific pretraining, improved tokenization, or dedicated architectural adaptations.

**Multilingual vs. English-Only Training** To evaluate the impact of multilingual data, we compared three models: the original ColQwen2-v0.1, and two ColQwen2 models trained from Qwen2-VL-2B using the training set from the original paper[3]. ColQwen2 (E) was trained only on this set, while ColQwen2 (M) was trained on the same set combined with our multilingual data. As shown in Table 3, including multilingual data improved performance across multiple languages compared to training with English data alone.

## 4.4 VISION QUESTION-ANSWERING RESULTS

We benchmark answer generation performance of three open-source MLLMs, Phi-4-multimodal Abouelenin et al. (2025), Paligemma2-3B Steiner et al. (2024), and InternVL2-4B Chen et al. (2023).

---

[3]Train Set: https://huggingface.co/datasets/vidore/colpali_train_set

| Accuracy | Figure | | Table | | Text | | Average | |
|---|---|---|---|---|---|---|---|---|
| | GAcc | PNLS | GAcc | PNLS | GAcc | PNLS | GAcc | PNLS |
| OpenAI-o3 (all) | 0.62 | 0.36 | **0.65** | 0.59 | **0.90** | 0.75 | **0.72** | 0.56 |
| GPT-4o (all) | 0.53 | 0.75 | 0.57 | **0.65** | 0.85 | **0.83** | 0.65 | **0.74** |
| OpenAI-o3 | 0.59 | 0.40 | **0.65** | 0.61 | 0.84 | 0.78 | 0.69 | 0.60 |
| Gemini-2.5 | 0.46 | 0.63 | 0.60 | 0.64 | 0.77 | 0.79 | 0.61 | 0.69 |
| GPT-4o | 0.48 | 0.59 | 0.55 | 0.64 | 0.84 | 0.82 | 0.62 | 0.68 |
| Paligemma2-3B | 0.03 | 0.02 | 0.11 | 0.18 | 0.43 | 0.43 | 0.19 | 0.21 |
| Phi-4-multimodal | 0.10 | 0.02 | 0.34 | 0.35 | 0.51 | 0.47 | 0.32 | 0.28 |
| InternVL2-4B (top 5) | **0.75** | **0.90** | 0.33 | 0.53 | 0.66 | 0.70 | 0.58 | 0.71 |

| Accuracy | Spanish | | Italian | | German | | French | | Dutch | | Arabic | | Croatian | | Japanese | |
|---|---|---|---|---|---|---|---|---|---|---|---|---|---|---|---|---|
| | GAcc | PNLS | GAcc | PNLS | GAcc | PNLS | GAcc | PNLS | GAcc | PNLS | GAcc | PNLS | GAcc | PNLS | GAcc | PNLS |
| OpenAI-o3 (all) | 0.87 | 0.74 | **0.79** | 0.68 | **0.84** | 0.69 | **0.86** | 0.76 | **0.82** | 0.71 | **0.79** | **0.62** | 0.83 | 0.70 | **0.76** | **0.62** |
| GPT-4o (all page) | **0.88** | **0.82** | 0.73 | **0.84** | 0.74 | 0.74 | **0.86** | **0.85** | 0.68 | 0.70 | 0.75 | 0.60 | **0.92** | **0.78** | 0.50 | 0.61 |
| OpenAI-o3 | 0.81 | 0.78 | 0.80 | 0.73 | **0.84** | 0.78 | 0.82 | 0.74 | 0.78 | 0.74 | 0.54 | 0.53 | 0.74 | 0.71 | 0.60 | 0.53 |
| Gemini-2.5 | 0.67 | 0.73 | 0.72 | 0.76 | **0.84** | 0.77 | 0.72 | 0.76 | 0.75 | **0.75** | 0.43 | 0.44 | 0.71 | 0.74 | 0.55 | 0.57 |
| GPT-4o | 0.77 | 0.81 | 0.70 | 0.80 | 0.80 | **0.80** | 0.73 | **0.85** | 0.73 | 0.74 | 0.39 | 0.58 | 0.71 | 0.67 | 0.50 | 0.51 |
| Phi-4-multimodal | 0.54 | 0.51 | 0.47 | 0.40 | 0.38 | 0.37 | 0.53 | 0.49 | 0.36 | 0.35 | 0.14 | 0.12 | 0.49 | 0.43 | 0.24 | 0.24 |
| Paligemma2-3B | 0.27 | 0.34 | 0.30 | 0.33 | 0.26 | 0.30 | 0.39 | 0.35 | 0.28 | 0.31 | 0.11 | 0.00 | 0.31 | 0.35 | 0.15 | 0.19 |
| InternVL2-4B (top 5) | 0.53 | 0.67 | 0.49 | 0.60 | 0.41 | 0.60 | 0.61 | 0.75 | 0.35 | 0.60 | 0.04 | 0.20 | 0.29 | 0.42 | 0.23 | 0.29 |

| Accuracy | Swedish | | Vietnamese | | Portuguese | | Finnish | | Czech | | Slovenian | | Danish | | Average | |
|---|---|---|---|---|---|---|---|---|---|---|---|---|---|---|---|---|
| OpenAI-o3 (all) | **0.84** | 0.67 | **0.90** | 0.67 | 0.85 | 0.69 | 0.71 | 0.67 | **0.81** | 0.57 | 0.94 | 0.69 | **1.00** | 0.79 | **0.84** | 0.68 |
| GPT-4o (all page) | 0.82 | **0.90** | 0.80 | **0.75** | 0.80 | **0.94** | **0.88** | **0.90** | 0.60 | **0.90** | **1.00** | **1.00** | **1.00** | **0.91** | 0.77 | **0.79** |
| OpenAI-o3 | 0.78 | 0.69 | 0.80 | 0.59 | **0.88** | 0.79 | 0.71 | 0.69 | **0.81** | 0.69 | 0.94 | 0.74 | 0.77 | 0.75 | 0.77 | 0.70 |
| Gemini-2.5 | 0.64 | 0.68 | 0.65 | 0.60 | 0.85 | 0.85 | 0.62 | 0.73 | 0.75 | 0.71 | 0.65 | 0.72 | 0.62 | 0.77 | 0.68 | 0.71 |
| GPT-4o | 0.70 | 0.73 | 0.70 | 0.63 | 0.85 | 0.82 | 0.71 | 0.80 | 0.69 | 0.70 | 0.75 | 0.77 | 0.69 | 0.82 | 0.70 | 0.74 |
| Phi-4-multimodal | 0.30 | 0.26 | 0.20 | 0.19 | 0.62 | 0.39 | 0.29 | 0.25 | 0.44 | 0.33 | 0.12 | 0.21 | 0.31 | 0.31 | 0.39 | 0.36 |
| Paligemma2-3B | 0.26 | 0.30 | 0.15 | 0.11 | 0.42 | 0.31 | 0.14 | 0.19 | 0.38 | 0.25 | 0.19 | 0.26 | 0.23 | 0.26 | 0.26 | 0.28 |
| InternVL2-4B (top 5) | 0.37 | 0.48 | 0.20 | 0.41 | 0.46 | 0.71 | 0.10 | 0.37 | 0.12 | 0.32 | 0.06 | 0.32 | 0.15 | 0.52 | 0.36 | 0.53 |

Table 4: Vision question-answering results on VisR-BenchEnglish split (upper), multilingual split (bottom). Bold font indicates the best overall performance for each language.

The results are reported in Table 4. For open-source models, we use ColPaliv1.2 to retrieve the most relevant page as the evidence page and perform inference on a single page. For GPT-4o, we include a baseline where the model is given all pages as input, providing an upper bound for retrieval-dependent models. Below are some findings that can be inferred from the experimental results.

**Multi-page models achieve strong performance.** Across all content types—except for figures—o3/GPT-4o (all) consistently obtains the highest GAcc and PNLS scores. This indicates that supplying the full document context substantially improves answer accuracy. In addition, o3 tends to be more semantically accurate, whereas GPT-4o shows stronger string-matching behavior, likely because both the questions and reference answers were generated by GPT-4o itself.

**Phi-4-multimodal and Paligemma2-3B perform poorly in most scenarios.** Phi-4-multimodal and Paligemma2-3B fail to provide reliable answers, with low GAcc and PNLs scores across all content types and languages. This suggests that models are not optimized for document VQA task.

**Multilingual performance varies significantly.** Although GPT-4o generally performs well across multiple languages, its performance drops in Arabic and low-resource languages (e.g., Croatian, Czech, Vietnamese, and Slovenian). This highlights the challenges of multilingual document retrieval, which further highlights the contribution of our VisR-Bench.

### 4.5 QUALITATIVE ERROR ANALYSIS

Due to space limitations, we include qualitative examples of retrieval failures and comparisons between true and hard negative pages in Appendix Figure A.1.

## 5 CONCLUSION

We introduce VisR-Bench, the first multilingual, question-driven visual retrieval benchmark for long documents, designed to evaluate retrieval performance across diverse document types and languages. Our evaluations across text-based retrieval methods, multimodal encoders, and MLLMs reveal that while MLLMs outperform other approaches, they still struggle with structured content and low-resource languages, exposing critical gaps in multilingual multimodal retrieval. By establishing a comprehensive evaluation framework, VisR-Benchpaves the way for future advancements in document-aware multimodal retrieval and RAG systems, advancing the way to more robust and linguistically diverse MLLM-based retrieval systems.

## 6 LANGUAGE MODEL USAGE STATEMENT

In preparing this manuscript, we used GPT-5 only for grammar checking and minor language polishing. The authors reviewed and edited all suggestions. All scientific content, analysis, and conclusions are entirely the work of the authors.

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

## A DEMO FIGURE-BASED RETRIEVAL

Question: What is the height of the door labeled as TYPE 5 in the image?
Answer: The height of the TYPE 5 door is 6'-8 1/2".

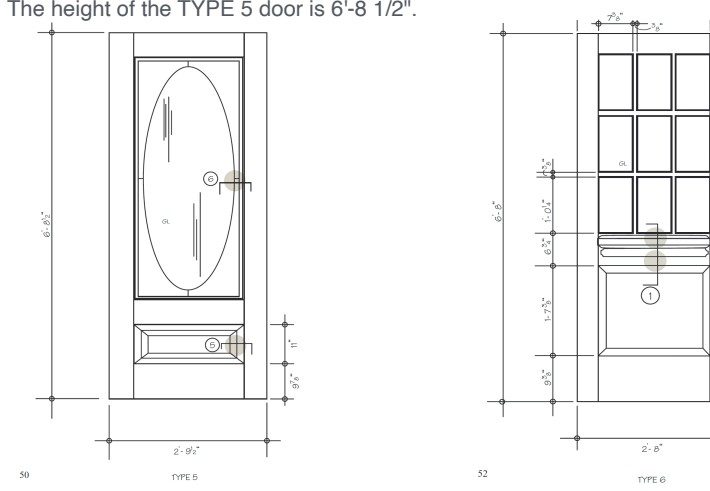

Positive Evidence                    Hard Negative

Question: Which hybrid combination shows the highest grain yield advantage, and what is its advantage value?
Answer: The hybrid combination Mon810+Mon836+Nk603 shows the highest grain yield advantage, with a value of 15 bu/A.

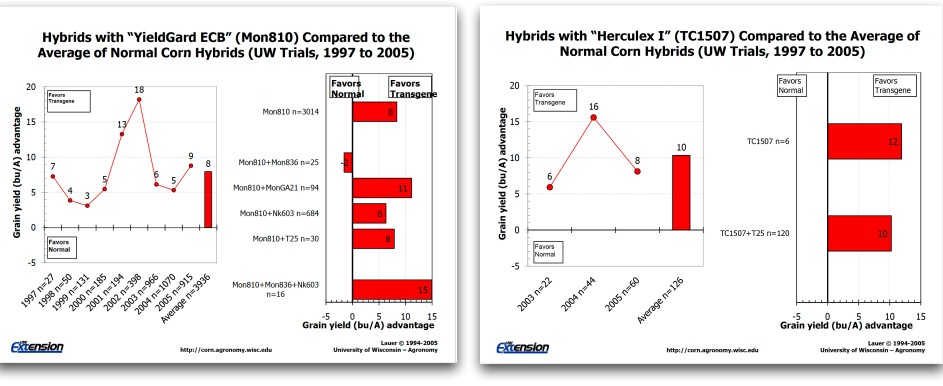

Positive Evidence                    Hard Negative

Figure A.1: **Qualitative error analysis for figure-based question answering.** This figure presents two examples where the GME model fails to retrieve the correct evidence page, while ColPali-v1.2 successfully identifies it. The incorrect pages retrieved by GME are shown as hard negatives. Notably, these hard negatives are highly similar to the correct evidence: in the top example, both pages are architectural blueprints containing references to door types and numbers such as "5" and "6"; in the bottom example, both figures depict grain yield advantage plots and contain the keyword "hybrid". These visually and semantically similar distractors demonstrate typical failure cases for GME and highlight the improved discriminative ability of ColPALI-v1.2 in retrieving the truly relevant figure.

## B  SYSTEM PROMPTS

### B.1  FIGURE-RELATED QUESTION

```
You are an AI assistant designed to refine and improve question-answer
pairs for document evidence retrieval. Your task is to enhance the given
QA pair by making the question self-contained and explicitly
identifying the relevant figure, table, or section within the document.

Guidelines:
 1. Clarify Ambiguous References
 • If the question refers to a figure, table, or section without specifying
    which one, revise it to include explicit identifiers (e.g., "Figure 3"
    or "the bar chart titled 'Sales Trends'").
 • Ensure the question contains enough context so that the audience can
    locate the evidence page without prior knowledge.
 2. Extract Contextual Cues
 • Use captions, labels, headings, or surrounding text to infer the most
    precise reference.
 • If multiple figures or tables exist, distinguish them based on their
    title, description, or content.
 3. Maintain Original Meaning
 • Preserve the intent and focus of the original question while making
    it self-contained.
 • Ensure clarity and specificity without adding unnecessary details.

Examples:

Input QA Pair (Ambiguous Question)
 • Q: What is the value of Data 3 in the chart?
 • A: The value of Data 3 is 45.

Refined QA Pair (Self-Contained Question)
 • Q: In Figure 5, which presents the monthly sales distribution, what is
    the value of Data 3 in the bar
    chart?
 • A: The value of Data 3 is 45.

Expected Output Format:

Output Format (JSON Response)
Generate a valid JSON response structured as follows:
{
    "detected_language": "LANGUAGE",
    "question_in_document_language": "XXXXXX",
    "question_in_english": "XXXXXX",
    "answer_in_document_language": "YYYYYY",
    "answer_in_english": "YYYYYY"
}

- Replace `LANGUAGE` with the detected language (e.g., "French", "Spanish").
- Replace `XXXXXX` with the generated questions.
- Replace ''YYYYYY'' with the corresponding answers extracted from the text.

Your response should include:
 1. The revised question that clearly specifies the evidence page.
 2. The original answer (unchanged, unless adjustments are necessary for clarity).
```

Now, please revise the given question pair according to these guidelines.

## B.2 TEXT-RELATED QUESTION

You are an assistant specialized in multilingual document retrieval tasks.

The task is as follows: given the text content of a document page, detect the language of the document and generate questions in the detected language as well as its English version.

Each question should:
1. The question should be relevant to the page, and should not be too general. The question should be about the subject of the page, and the answer needs to be found in the page.
2. The question is asked by a user to get information from a multi-page document. Generate a question that could be asked by provided infomation in the given page.
3. Generate as well the answer to the question, which should be found in the page.

Please do not generate:
1. Questions that are too broad or global (e.g., summarization or conclusion-type questions that
require information beyond the given page).
2. Questions that are not specific to the page (e.g., questions that apply equally to all pages, like
"What is the page number?").
3. Questions that require cross-page reasoning or involve multiple pages to answer.

For each question:
- Generate its corresponding answer, which must be found explicitly in the text content of the page.
- The answer should be formatted as words or phrases extracted directly from the text.
- Questions and answers must be provided in both the detected language and in English.

Generate at most THREE pairs of questions and answers per page in a dictionary format. If no relevant questions can be generated for the page, return an empty list. The output format should include only a valid json string, such as:
```
{
    "detected_language": "LANGUAGE",
    "questions": [
        {
            "question_in_detected_language": "XXXXXX",
            "question_in_english": "XXXXXX",
            "answer_in_detected_language": "YYYYYY",
            "answer_in_english": "YYYYYY"
        },
        {
            "question_in_detected_language": "XXXXXX",
            "question_in_english": "XXXXXX",
            "answer_in_detected_language": "YYYYYY",
            "answer_in_english": "YYYYYY"
        },
        {
```

```
            "question_in_detected_language": "XXXXXX",
            "question_in_english": "XXXXXX",
            "answer_in_detected_language": "YYYYYY",
            "answer_in_english": "YYYYYY"
        }
    ]
}
```

- Replace `LANGUAGE` with the detected language (e.g., "French", "Spanish").
- Replace `XXXXXX` with the generated questions.
- Replace `'YYYYYY'` with the corresponding answers extracted from the text.
- If no questions can be generated, return an empty list.

Focus on crafting meaningful and diverse questions that represent realistic user queries about the document.

Here is the text:

### B.3 TABLE-RELATED QUESTION

You are an intelligent assistant specialized in multilingual document analysis and table-based question generation.
Your goal is to generate computational or reasoning-based questions from a given interleaved document page.

Task Overview
Given the content of a document page (text and tables), you must:
1. Detect the language of the document.
2. Generate at most three pairs of questions and answers, where:
   - Questions should require reasoning, computation, or trend analysis (not direct lookups).
   - Each question must be generated in both the detected language and English.
   - Answers must be extracted from the table in the page and formatted as words or phrases.

Question Requirements
Require computation or logical inference rather than simple fact lookup.
Analyze trends, comparisons, rankings, or correlations in the table data.
Ensure relevance to the page while avoiding overly general or document-wide questions.
The question should require information from the table to answer.

Example Question Types
Trend Analysis: "How has X changed over the last five years?"
Growth Rate: "Which category experienced the fastest increase?"
Comparison: "Which product had the highest price difference between regions?"
Correlations: "Does an increase in X correspond to a decrease in Y?"

Restrictions: Do Not Generate
Fact-based questions (e.g., "What is the value of X in row 3, column 2?").
Broad summarization or conclusions beyond the given page.
Questions requiring multi-page reasoning.
Irrelevant metadata questions (e.g., "What is the page number?").

Output Format (JSON Response)
Generate a valid JSON response structured as follows:
```
{
    "detected_language": "LANGUAGE",
```

```
918        "questions": [
919            {
920                "question_in_detected_language": "XXXXXX",
921                "question_in_english": "XXXXXX",
922                "answer_in_detected_language": "YYYYYY",
923                "answer_in_english": "YYYYYY"
924            },
925            {
926                "question_in_detected_language": "XXXXXX",
927                "question_in_english": "XXXXXX",
928                "answer_in_detected_language": "YYYYYY",
929                "answer_in_english": "YYYYYY"
930            },
931            {
932                "question_in_detected_language": "XXXXXX",
933                "question_in_english": "XXXXXX",
934                "answer_in_detected_language": "YYYYYY",
935                "answer_in_english": "YYYYYY"
936            }
937        ]
938    }

939    - Replace `LANGUAGE` with the detected language (e.g., "French", "Spanish").
940    - Replace `XXXXXX` with the generated questions.
941    - Replace `'YYYYYY'` with the corresponding answers extracted from the text.

942    Focus on designing meaningful and diverse questions that reflect realistic
943    user queries and require information from the table.

945    Here is the text:
```

B.4  GENERAL MULTILINGUAL QUESTION

```
You are an assistant specialized in multilingual document retrieval tasks.

The task is as follows: given the text content and image content of a document
page, detect the language of the document and generate questions in the
detected language as well as its English version.

Each question should:
1. The question should be relevant to the page, and should not be too
specific or too general. The question should be about the subject of the
page, and the answer needs to be found in the page.
2. The question is asked by a user to get some information from a large
documentary corpus that contains multimodal data.
Generate a question that could be asked by a user without knowing
the existence and the content of the corpus.
3. Generate as well the answer to the question, which should be found in the
page. And the format of the answer should be a list of words answering the
question.

Please do not generate:
1. Questions that are too broad or global
(e.g., summarization or conclusion-type questions that require information
beyond the given page).
2. Questions that are not specific to the image
(e.g., questions that apply equally to all pages, like "What is the page
number?").
3. Questions that require cross-page reasoning or involve multiple pages
```

```
to answer.

For each question:
- Generate its corresponding answer, which must be found explicitly in the
text content of the page.
- The answer should be formatted as a list of words or phrases extracted
directly from the text.
- Questions and answers must be provided in both the detected language
and in English.

Generate at most THREE pairs of questions and answers per page in a
dictionary format.
If no relevant questions can be generated for the page, return an empty list.
The output format should include only a valid json string, such as:
{
    "detected_language": "LANGUAGE",
    "questions": [
        {
            "question_in_detected_language": "XXXXXX",
            "question_in_english": "XXXXXX",
            "answer_in_detected_language": "YYYYYY",
            "answer_in_english": "YYYYYY"
        },
        {
            "question_in_detected_language": "XXXXXX",
            "question_in_english": "XXXXXX",
            "answer_in_detected_language": "YYYYYY",
            "answer_in_english": "YYYYYY"
        },
        {
            "question_in_detected_language": "XXXXXX",
            "question_in_english": "XXXXXX",
            "answer_in_detected_language": "YYYYYY",
            "answer_in_english": "YYYYYY"
        }
    ]
}

- Replace `LANGUAGE` with the detected language (e.g., "French", "Spanish").
- Replace `XXXXXX` with the generated questions.
- Replace ''YYYYYY'' with the corresponding answers extracted from the text.
- If no questions can be generated, return an empty list.

Focus on crafting meaningful and diverse questions that represent realistic
user queries about the document.

Here is the text:
```

