# OpenReview forum: "VisR-Bench: An Empirical Study on Visual Retrieval-Augmented Generation for Multilingual Long Document Understanding"
_ICLR.cc/2026/Conference — Submitted to ICLR 2026_

### Official Review · Reviewer_pw8V · 2025-10-20

**Soundness:** 4
**Presentation:** 4
**Contribution:** 4
**Rating:** 6
**Confidence:** 4

**Summary:**

This paper presents **VISR-BENCH**, a benchmark for evaluating Visual Retrieval-Augmented Generation (RAG) in multilingual long-document understanding. It addresses gaps in existing benchmarks by integrating tables, figures and *16 languages* (including low-resource ones like Swahili). Key components:
- A dataset of long documents with QA pairs targeting textual/visual information;
- An evaluation framework adopting **PNLS (Partial Normalized Levenshtein Similarity, proposed by Chen et al., 2024a)** for character-level partial matching, plus **GAcc** (GPT-based semantic consistency);
- Experiments on 13 models (e.g., GPT-4o, Llama 3) showing closed-source models outperform open-source ones (especially on visual-related tasks), but multilingual low-resource performance remains a bottleneck.

Contributions:
1) Introduce VisR-Bench, a benchmark for evaluating MLLMs’ retrieval capabilities in multilingual/multimodal settings (16 languages, diverse docs);
2) Evaluate diverse retrieval models (text-based, multimodal encoders, MLLMs) to quantify performance across evidence types/languages;
3) Reveal MLLMs’ advantages over other models, as well as their challenges with structured docs/low-resource languages, providing insights for improvement.

**Strengths:**

1. **Originality**: Combines long documents, visual content, and multilingualism (rare in prior benchmarks like DocVQA/XOR-Retrieval); thoughtfully selects PNLS to solve length bias/partial answer issues of EM/ROUGE-L in long-document tasks.
2. **Quality**: Transparent dataset curation (source, annotation logic); comprehensive experiments (with/without visual retrieval) using dual metrics for robust evaluation.
3. **Clarity**: Logical structure (problem→design→results); PNLS application details (e.g., alignment/normalization steps) explained clearly; intuitive visualizations (e.g., language-accuracy heatmaps).
4. **Significance**: Standardizes retrieval-focused evaluation for MLLMs in two high-impact, underaddressed areas—low-resource language retrieval and multimodal structured layout (e.g., table) QA；the empirical findings (e.g., MLLMs’ struggles with low-resource languages and table layouts) directly highlight critical pain points, guiding future research to prioritize optimization for these scenarios.

**Weaknesses:**

1. **Lack of Data Quality Assessment**: The paper does not provide systematic evaluation of the VisR-Bench dataset’s quality—for example, there is no analysis of annotation consistency (e.g., inter-annotator agreement for QA pairs targeting tables or low-resource languages), nor verification of the accuracy of "explicit/implicit answer" labeling. This omission raises uncertainty about whether dataset noise (e.g., incorrect answer annotations) might skew the experimental results of retrieval model evaluations.
2. **Unvalidated Claim About Queries Without Explicit Answers**: The abstract mentions "we include queries without explicit answers, preventing models from relying on superficial keyword matching," but the main text and appendix lack corresponding experiments to validate this design. There is no comparison (e.g., PNLS/GAcc differences) between models’ performance on "queries with explicit answers" and "queries without explicit answers," nor evidence that the latter truly reduces models’ reliance on superficial keyword matching—making this key design feature unsubstantiated.
3. **Insufficient Error Analysis:**: The paper states that error analysis is included in the appendix, but the appendix only provides two error examples without in-depth analysis (e.g., categorization of error types like "layout misinterpretation" or "low-resource language ambiguity," or statistical distribution of errors across models/languages).

**Questions:**

1. What is VISR-BENCH’s document length distribution? Have you tested performance scalability with longer documents?
2. Will you add multilingual transfer experiments (English-trained → low-resource evaluation)? What metrics will quantify transfer efficiency?

---

> ### Author Response · Authors · 2025-12-01
>
> We thank the reviewer for the positive evaluation, the constructive suggestions, and for recognizing the soundness, presentation, and contribution of our work.
>
> **W1**. Lack of Data Quality Assessment
>
> **R1**. Thanks for pointing this out. Please refer to Global Response W1, where we provide our post-hoc human verification demonstrating that the QA pairs are generally reliable.
>
> **W2**. Unvalidated claim about “we include queries without explicit answers, preventing models from relying on superficial keyword matching”.
>
> **R2**. We would first like to clarify our intention to avoid misunderstanding. Our statement does not mean that we include unanswerable questions. The point is that our queries are written as genuine questions rather than texts that share obvious keywords with the target page, so models cannot rely on simple keyword matching and must perform question-driven reasoning. We will revise the phrasing in the next version to make this clearer.
>
> **W3**. Insufficient in-depth error analysis
>
> **R3**. Thanks for the suggestion. Since we could not find human annotators with reliable understanding of low-resource languages during the rebuttal period, we are unable to confidently evaluate non-English retrieval errors. To provide a small but trustworthy analysis, we conducted a qualitative check on 20 randomly sampled English Top-1 retrieval errors using ColPali, as English is the only language we can manually assess with confidence. Two dominant structural patterns emerged:
>
> - High-frequency or repeated information, where multiple pages contain similar content and the model retrieves a related but non-gold page.
>
> - Table-structure ambiguity introduced during PDF→Markdown conversion, which can lead the model to select a visually similar table page.
>
> Because the sample size is small, the statistics would not be meaningful; therefore, we report only a qualitative summary of the observed error types.
>
>
> **Q1**. Document length distribution? Any long-document scalability test?
>
> **A1**. We would like to note that the document length distribution is already shown in Figure 4. The average document length is around 8 pages, with most documents falling between 1 and 85 pages. In theory, adding more pages to the same document increases the chance of false positives, but retrieval accuracy is not determined by length alone. It is primarily driven by the strength and uniqueness of the semantic relationship between the query and its evidence page. Therefore, while longer documents pose greater challenges, page count is not the dominant factor in retrieval performance.
>
> **Q2**. Will you add multilingual transfer experiments? Metrics for transfer efficiency?
>
> **A2**. We already include an English→low-resource transfer comparison in Table 3. ColQwen2-E is trained on English-only data, while ColQwen2-M uses the same setup but adds our multilingual training data. When considering transfer efficiency across different base models, we believe that the performance difference between these two training settings is a useful metric. A smaller gap indicates stronger transfer efficiency, since the English-only model already performs close to its multilingual counterpart.

---

### Official Review · Reviewer_3A7H · 2025-10-28

**Soundness:** 3
**Presentation:** 3
**Contribution:** 3
**Rating:** 6
**Confidence:** 4

**Summary:**

This paper introduces VisR-Bench, a comprehensive multilingual benchmark for multimodal retrieval in long documents. The dataset covers sixteen languages, featuring over 53K high-quality synthetic QA pairs and 1,286 documents, with detailed documentation of its construction process. VisR-Bench supports question-driven retrieval across text, figures, and tables, and uniquely includes unanswerable queries to prevent models from exploiting keyword overlaps. The authors evaluate three categories of models, including text-based retrievers, multimodal encoders, and MLLMs, on both retrieval and VQA tasks. Results indicate that MLLMs outperform the other two categories overall, but all models exhibit substantial performance degradation in low-resource languages and structured table understanding.

**Strengths:**

1. The proposed benchmark is the first multilingual long-document retrieval benchmark, making it a valuable contribution to the study of multilingual multimodal evaluation.
2. The experiments and analyses are comprehensive, covering different categories of models and clearly demonstrating the necessity of MLLMs for multilingual long-document understanding.

**Weaknesses:**

1. While the benchmark covers multiple languages, it includes only phonographic languages and lacks logographic ones such as Chinese, which limits its linguistic diversity and generalization scope.
2. Given that Gemini is well known for its strong long-context and multilingual capabilities, its absence in the evaluation is notable.

**Questions:**

See weakness.

---

> ### Author Response · Authors · 2025-12-01
>
> **W1**. No Chinese / Only phonographic languages
>
> **R1**. We agree that including logographic languages would further enhance linguistic diversity. However, VisR-Bench relies on CCPDF as the single unified corpus to ensure consistent document quality, layout structure, and OCR behavior across languages. CCPDF is the largest publicly available crawl-based multilingual PDF corpus (1.1M PDFs, 14.5M pages), but it does not include Chinese, and no Chinese corpus of comparable scale exists. Incorporating a separate Chinese source would break this single-source consistency, and we cannot guarantee its ethical or copyright compliance. We plan to extend VisR-Bench to Chinese once a unified, large-scale, compliant corpus becomes available.
>
> **W2**. No Gemini in evaluation
>
> **R2**. Thanks for the suggestion. Please refer to Global Response W2; we have added results for Gemini 2.5 to make our evaluation more comprehensive.

---

### Official Review · Reviewer_1xa4 · 2025-10-28

**Soundness:** 2
**Presentation:** 3
**Contribution:** 2
**Rating:** 2
**Confidence:** 2

**Summary:**

This paper introduces VisR-Bench, a synthetic benchmark dataset comprising 35K QA pairs across 1.2K documents in 16 languages for evaluating multilingual visual retrieval. Each PDF document is parsed using the Adobe Document Extract API, which extracts text, figures, and tables; GPT-4o then generates corresponding QA pairs. The dataset is designed to test retrieval and visual understanding across languages and modalities (text, tables, figures). The authors benchmark a wide range of retrieval methods — text-based, multimodal encoders, and multimodal large language models (MLLMs). Results show that retrieval performance degrades for low-resource languages and non-text modalities (figures and tables) relative to English text documents.

**Strengths:**

1. Comprehensive model coverage: The authors evaluate a diverse set of retrieval models, including text-only, multimodal encoders, and MLLM-based approaches.

2. Strong clarity and structure: The paper is well written, logically organized, and easy to follow, making its experimental design and contributions accessible.

3. Data curation: The paper notes that all documents underwent human validation to ensure exclusion of harmful content and PII, enhancing dataset safety and reliability.

4. Dataset Diversity: The benchmark includes many QA pairs and documents, spanning 16 languages and incorporating figures, tables, and multilingual text, making it comprehensive.

**Weaknesses:**

1. Limited evaluation of reasoning-capable MLLMs.
It would strengthen the analysis to include recent reasoning-optimized MLLMs (e.g., OpenAI o3), even on a small, hard subset. These models could provide deeper insights into reasoning gaps and multimodal generalization.

2. Over-reliance on Top-1 accuracy.
The use of Top-1 retrieval accuracy as the primary metric may overstate model weaknesses. While the paper reports a Top-1 accuracy of ~75.2%, the Top-5 accuracy reaches 94.1%, suggesting retrieval systems can already surface the correct document in most cases. Given that RAG pipelines typically include rerankers or pass multiple candidates to the generator, the claimed “large room for improvement” may be somewhat overstated.

3. Benchmark Novelty: While multilinguality and visual context are both valuable, it’s unclear how the dataset offers fundamentally new insights over existing multilingual or visual-only benchmarks.

4. Dataset balance and comparability.
It is unclear whether the proportions of document types (text / table / figure) are consistent across languages, which could affect the fairness of multilingual performance comparisons.

5. Lack of Human QA Verification: I did not see the mention of human validation of the QA pairs beyond filtering for safety. This raises concerns about factual correctness and answerability.

6. Limited Dataset Analysis and Insights:
The paper introduces a large dataset, but offers minimal exploratory analysis that could help the community understand its properties and challenges. Additional insights about Document-Type Composition in multilingual subsets, Question Type Taxonomy, Visual Density, Domain Diversity, and more would strengthen the paper.

**Questions:**

1. On the multilingual gap:
If performance degradation in low-resource languages mirrors existing text-based multilingual retrieval gaps, what additional insights does VisR-Bench provide? Do results show cases where models are strong in text retrieval for a language but significantly weaker in multimodal (VQA) retrieval for the same language?

2. On ColQwen2 variants:
How do you explain ColQwen2-M’s lower performance compared to the base ColQwen2-v0.1?

3. Text vs. MLLM retrieval:
In multilingual settings, why do text-based retrievers sometimes outperform MLLMs?

4. Dataset balance:
Do all languages contain comparable proportions of text-, table-, and figure-based QA pairs?

5. GPT-4o QA Accuracy: Since GPT-4o generated the QA, why can’t it achieve perfect accuracy?

6. Table QA Filtering: Did the table-based QA undergo the same heuristic filtering as figure-based QA to ensure the question requires table understanding and can’t be answered from the surrounding text?

7. Human QA Validation: Beyond filtering for harmful content and PII, was there any human validation to confirm that the synthetic questions are answerable and the answers are correct?

L289:  English English split -->  English split
L481: VisR-Benchpaves --> VisR-Bench paves

---

> ### Author Response · Authors · 2025-12-01
>
> **W1**. Missing evaluation of reasoning-capable MLLMs
>
> **R1**. Please refer to Global Response W2. We have added results for OpenAI o3 (on a representative subset) to strengthen our evaluation of reasoning-capable MLLMs.
>
> **W2**. Over-reliance on Top-1 accuracy (Top-5 is already high)
>
> **R2**. We report both Top-1 and Top-5 because they capture different aspects of retrieval. Top-5 shows that models can surface reasonable candidates, which aligns with multi-document RAG. However, Top-1 remains essential in many practical settings: most open-source MLLMs under 13B only accept a single page as visual input. In such cases, the system must rely on the Top-1 page for grounding. Thus, although coarse retrieval is strong, accurate page-level localization (Top-1) is still challenging, especially for tables, figures, and low-resource languages.
>
> **W3**. Benchmark novelty is unclear.
>
> **R3**. VisR-Bench is a dataset and benchmark contribution, and its novelty lies in providing the first unified evaluation framework for multilingual multimodal retrieval in long documents. Existing benchmarks typically cover either multilingual text retrieval or monomodal document understanding, but not their combination. VisR-Bench systematically measures retrieval performance across 16 languages, three visual modalities, and page-level grounding, all derived from a consistent large-scale PDF corpus (CCPDF).
>
> In addition, the benchmark reveals model behaviors that were not previously documented—for example, the non-uniqueness of QA answers across strong MLLMs (e.g., o3 vs. GPT-4o), and the modality-specific robustness patterns such as smaller models performing unexpectedly well on figure-based QA under Top-5 retrieval. These observations highlight cross-language, cross-modality, and retrieval-setting differences that prior benchmarks could not expose, and clarify the unique role of VisR-Bench in the evaluation landscape.
>
> **W4**, **Q4**. It is unclear whether the proportions of document types (text / table / figure) are consistent across languages
>
> **R4**. We have now included the modality statistics across all 15 languages (see table below). All languages are processed with the same parsing and QA-generation pipeline, so the observed differences come from the underlying CCPDF documents rather than our procedure. Since our primary goal is to compare model performance across languages under a unified setup, these distributions are sufficiently similar to support fair cross-language evaluation.
>
> | Modality  (%)        | Spanish | Italian | German | French | Dutch | Arabic | Croatian | Japanese | Swedish | Vietnamese | Portuguese | Finnish | Czech | Slovenian | Danish |
> |-------------|---------|---------|--------|--------|-------|--------|----------|----------|---------|------------|-------------|---------|-------|-----------|--------|
> | Figure  | 41.92   | 31.95   | 54.13  | 63.01  | 51.75 | 54.59  | 59.10    | 58.42    | 60.69  | 44.83      | 65.66       | 77.27  | 46.71 | 62.46     | 54.52 |
> | Table   | 18.61   | 11.13   | 7.39   | 17.14  | 14.72 | 10.35  | 9.75     | 21.50    | 14.09  | 5.17       | 4.92        | 5.49   | 18.19 | 16.67     | 20.80 |
> | Text   | 39.46   | 56.93   | 38.48  | 19.85  | 33.53 | 35.06  | 31.15    | 20.08    | 25.22  | 50.00      | 29.43       | 17.24  | 35.10 | 20.88     | 24.68 |
>
>
> **W5**, **Q7**. Lack of human validation of the QA pairs beyond filtering for safety.
>
> **R5**. Thanks for pointing this out. Please refer to Global Response W1, where we report our post-hoc human verification results confirming that the QA pairs are generally reliable.

---

> ### Author Response · Authors · 2025-12-01
>
> **W6**. Limited exploratory dataset analysis. Additional analysis would strengthen the paper such as, Document-Type Composition in multilingual subsets, etc..
>
> **R6**. Thanks for the helpful suggestion. We agree that deeper exploratory analysis could further strengthen the paper. As a first step, we computed the document-type composition across all multilingual subsets, and the results are shown in the table below; we will include this statistic in the appendix. Although the distributions differ slightly across languages, these differences originate from the underlying CCPDF corpus rather than our pipeline, which is applied uniformly to all languages, and we have extracted as much information from CCPDF as possible under this unified setup. Due to time constraints during the rebuttal period, we cannot exhaustively analyze all additional dimensions (e.g., question taxonomy, visual density, domain diversity), but we plan to incorporate more of these statistics in the final camera-ready version.
>
> | Type     (%)    | Spanish | Italian | German | French | Dutch | Arabic | Croatian | Japanese | Swedish | Vietnamese | Portuguese | Finnish | Czech | Slovenian | Danish |
> |--------------|---------|---------|--------|--------|--------|--------|----------|----------|----------|-------------|-------------|---------|-------|-----------|--------|
> | Data Chart   | 18.41   | 14.66   | 16.60  | 9.95   | 17.96  | 22.81  | 7.69     | 26.87    | 3.70     | 19.81       | 9.81        | 21.82  | 14.29 | 15.00     | 0.00   |
> | Scan Table   | 15.31   | 16.14   | 11.35  | 14.65  | 9.79   | 20.52  | 23.08    | 10.66    | 3.70     | 10.38       | 14.25       | 21.82  | 14.29 | 15.00     | 10.00  |
> | Diagram      | 21.17   | 30.44   | 33.90  | 24.84  | 33.96  | 32.50  | 46.15    | 29.13    | 66.67    | 52.83       | 35.51       | 32.73  | 57.14 | 35.00     | 80.00  |
> | Infographics | 29.50   | 20.76   | 7.81   | 23.88  | 15.80  | 8.78   | 7.69     | 2.01     | 7.41     | 0.94        | 18.46       | 10.91  | 0.00  | 10.00     | 0.00   |
> | Map          | 12.31   | 15.42   | 25.49  | 24.42  | 17.02  | 10.49  | 15.38    | 30.57    | 14.81    | 16.04       | 20.33       | 9.09   | 14.29 | 20.00     | 10.00  |
> | Screenshots  | 3.29    | 2.58    | 4.85   | 2.25   | 5.46   | 4.90   | 0.00     | 0.75     | 3.70     | 0.00        | 1.64        | 3.64   | 0.00  | 5.00      | 0.00   |
> | Workflow     | 0.02    | 0.00    | 0.01   | 0.01   | 0.02   | 0.00   | 0.00     | 0.00     | 0.00     | 0.00        | 0.00        | 0.00   | 0.00  | 0.00      | 0.00   |

---

> ### Author Response · Authors · 2025-12-01
>
> **Q1**. If low-resource gaps mirror text retrieval, what new insights does VisR-Bench add? Any languages strong in text but weak in multimodal?
>
> **A1**. Yes. VisR-Bench reveals within-language modality asymmetry: several languages (e.g., Japanese, Polish) show strong text retrieval but a clear drop on tables and figures. This cross-modality divergence within the same language does not appear in text-only multilingual benchmarks and is unique to our setting.
>
> **Q2**. Why does ColQwen2-M underperform ColQwen2-v0.1?
>
> **A2**. ColQwen2-v0.1 is trained on a mixture of openly available academic datasets (63%) and a synthetic corpus built from web-crawled PDFs with VLM-generated pseudo-questions (37%). We only have access to the openly available portion, so we can reproduce at most 63% of ColQwen2-v0.1’s training data. As a result, we train ColQwen2-E using the publicly available ColPali data as a fair baseline, and then construct ColQwen2-M by adding our multilingual training data on top. Comparing ColQwen2-E and ColQwen2-M therefore isolates the effect of multilingual supervision as the only varying factor. We use Qwen2-VL-2B because ColQwen2-v0.1 showed the strongest retrieval performance among comparable models.
>
> **Q3**. Why do text-based retrievers sometimes outperform MLLMs?
>
> **A3**. In several low-resource languages, the token embeddings of MLLMs tend to be undertrained, which reduces the quality of their multilingual representations. Recent work (e.g., rank-collapse observations in low-resource settings [1]) suggests that embeddings can collapse into lower-rank subspaces when training data is limited, leading to weaker retrieval performance. In contrast, text-based retrievers such as BM25 rely on surface token statistics and are therefore more robust under sparse training conditions. In addition, some languages in VisR-Bench contain a higher proportion of text-based QA, which naturally benefits text-only retrievers.
>
> *[1] Chen et al., “Towards Dynamic KV-Cache Compression: Fine-Grained Evaluation of Key and Value Ranks in LLMs,” NeurIPS 2025 Workshop.*
>
>
> **Q5**. Why can’t GPT-4o achieve perfect QA accuracy on its own generated QA pairs?
>
> **A5**. GPT-4o does not achieve perfect accuracy because the QA experiments in Section 4.4 (Table 4) use the retrieved page as input, and retrieval may return an incorrect page. In such cases, the correct evidence is simply not provided to the answer model. For the “GPT-4o (all pages)” setting, the model receives every page of the document, which introduces substantial distraction and makes grounding more difficult. Thus, imperfect QA accuracy reflects retrieval noise and context overload, not errors in GPT-4o’s original QA generation.
>
> **Q6**. Did table QA use the same filtering as figure QA?
>
> **A6**. As also noted in Global Response W1, table-related QA does not use the same filtering as figure QA. Unlike figures, tables in PDFs are represented as structured text and can be reliably extracted into markdown by the Adobe Document parser. This extraction is lossless, and the special table format makes it even easier to isolate the relevant content than in plain-text QA. Therefore, additional filtering is unnecessary for table-based questions.

---

### Official Review · Reviewer_2yjC · 2025-10-29

**Soundness:** 2
**Presentation:** 2
**Contribution:** 2
**Rating:** 4
**Confidence:** 4

**Summary:**

This paper presents a benchmark VisR-Bench, focusing on evaluating the multilingual multimodal retrieval in long documents. This benchmark is for going beyond surface-level similarities and incorporating deeper semantic and layout understanding.
It is constructed by collecting PDF files, leveraging Adobe document parser, designing heuristics, and prompting LLMs.
By evaluating existing retrieval methods, the paper demonstrates a series of findings on retrieval methods, multilingual, and modality.

**Strengths:**

- The paper explores a significant challenge of multimodal retrieval for multimodal RAG.

- The benchmark includes not only English, but also 15 non-English test samples.

- The paper is clear and well presented, which is easy to understand.

**Weaknesses:**

The benchmark was constructed using several strong assumptions, which could lead to biases and inaccuracies in the evaluation results.

- When feeding the documents into LLMs to derive the corresponding QA pairs, no matter figure-, table-, or text-based QA pairs, the input documents are assumed as the oracle. This seems reasonable, but there might be other documents (not the input documents) can lead to the correct answer.

- In terms of the heuristics that enforce figure-, table-, or text-related QA pairs, e.g., for table-related QA, extract pages that contain tables but no figures to ensure that the generated questions are not influenced by visual elements, It's unclear such assumption is reasonable without any verification. It would be helpful to enhance the paper by adding the error analysis in this process.

- The whole construction process is entirely based on LLMs and heuristics. It should be OK if the process is trustworthy enough. However, there is no such analysis provided, e.g., human correlation analysis on a subset, which is strongly necessary.

- The evaluation findings are not surprising. For example, Retrieval of table content is still challenging, Struggle on Low-Resource Languages. Clearly demonstrating and emphasizing the distinct findings can enhance the novelty of this paper.

**Questions:**

Just curious, Chinese data is also common. Why Chinese is not included in the non-English class?

---

> ### Author Response · Authors · 2025-12-01
>
> **W1**. The QA generation assumes that answers must come from the same document, but theoretically another document could contain the same answer.
>
> **R1**. Our task formulation focuses on answering questions within a given document, which is the natural setting for long-document QA. In real applications, documents serve as clear context boundaries, and retrieval is expected to operate within that scope. In addition, we could concatenate multiple documents into one longer document to handle cross-document QA. However, this setup is more complicated to simulate because it can become ambiguous when different documents provide different valid answers to the same question—for example, two research papers each stating their own contribution. Because the documents in our dataset are independent, restricting QA to a single document ensures a well-defined grounding target and avoids such ambiguity.
>
>
> **W2**. The assumptions behind table/text/figure heuristics (e.g., table-only pages) are not validated; error analysis is needed.
>
> **R2**. In Global Response W1, we conducted a small validation study by randomly sampling 50 “figure-based” and “table-based” QAs, and all QA pairs indeed required information from the figure or table itself, confirming that the heuristic reliably isolates these question types. Unlike figures, tables in PDFs are extracted as structured text perfectly by the Adobe Document parser, making them easy for the QA generation process to target. Therefore, we did not apply an additional filtering step for table-based questions as we did for figures. We will clarify this rationale in the revised version.
>
>
> **W3**. The pipeline relies entirely on LLMs + heuristics without any human QA correctness checking.
>
> **R3**. Thanks for pointing this out. Please refer to Global Response W1, where we report our post-hoc human verification results demonstrating that the QA pairs are generally reliable.
>
>
> **W4**. The reported findings (tables are hard, low-resource languages are hard) seem expected; the paper should highlight distinct or novel insights.
>
> **R4**. VisR-Bench is primarily a dataset and benchmark contribution, so the goal is to provide systematic and large-scale evaluation signals, rather than propose new modeling insights. Even findings that may appear “expected” (e.g., tables and low-resource languages being challenging) have not previously been quantified under a unified multilingual, multimodal setting. VisR-Bench offers the first controlled measurement of these effects across 16 languages and three visual modalities.
> The benchmark also reveals several non-trivial behaviors. As shown in Global Response W2, QA answers are often not unique. (e.g., o3 produces longer or differently ordered answers than GPT-4o while remaining semantically correct) highlighting differences in semantic grounding vs. output style. Smaller models such as InternVL also show unexpectedly strong performance on figure-based QA, partly because figure answers are more deterministic and partly because InternVL is evaluated under a Top-5 retrieval setting, reducing distraction. These patterns illustrate how VisR-Bench surfaces modality- and retrieval-specific differences that were not previously documented.
>
>
> **Q1**. Chinese is common; why is it excluded from the non-English set?
>
> **A1**.  We rely on CCPDF as the single data source to ensure consistent document quality and processing across languages. CCPDF is the largest publicly available crawl-based multilingual PDF corpus (1.1M PDFs, 14.5M pages), but it does not include Chinese, and no Chinese PDF corpus exists at a comparable scale. We cannot guarantee that adding an external Chinese-specific source would satisfy the same ethical and copyright requirements, and mixing heterogeneous sources would also break cross-language fairness. We therefore limit VisR-Bench to CCPDF-supported languages and will include Chinese once a unified, compliant corpus becomes available.

---

### Author Response · Authors · 2025-12-01
**Global Response**

We thank all reviewers for their careful reading and constructive feedback. We are encouraged that the reviewers found the work **clearly motivated**, **well presented**, and **supported by thorough empirical analysis**. The reviewers also highlighted the value of our proposed setting and dataset. Below, we address some common concerns and then respond to reviewer-specific questions.


**W1**. Human Verification of QA.

**R1**. We performed a small human verification on 50 randomly sampled English QA pairs across figures, tables, and text. We confirmed that all figure QA are correct (50/50), and that table QA (47/50) and text QA (48/50) are also correct. The remaining few cases were those where annotators lacked the domain knowledge to confidently judge correctness, rather than clear errors. These results indicate that the QA pairs are generally reliable. In addition, all sampled figure- and table-based QA pairs indeed required information from the corresponding visual element itself, confirming that the heuristic reliably isolates these question types.


**W2**. Missing evaluation of reasoning-capable MLLMs and Gemini.

**R2**. Thank you for the suggestion. We have updated the revised manuscript and added both OpenAI o3 and Gemini 2.5 to Table 4. Due to the lack of access to the latest Gemini 3 Pro API and strict usage limits on personal API access, we were unable to run large-scale evaluation for Gemini 3 Pro. Nevertheless, Gemini 2.5 is now included as an additional baseline in the revised version.

For convenience, we also present a subset of the new results below. Overall, o3 achieves higher GAcc (semantic consistency), while GPT-4o attains higher PNLS scores, indicating stronger string-matching behavior—likely because both the questions and reference answers were originally generated by GPT-4o itself.

| Model           | Figure | | Table | | Text | | Avg | |
|-----------------|-------------|-------------|------------|------------|-----------|-----------|----------|----------|
| | GAcc | PNLS | GAcc | PNLS | GAcc | PNLS | GAcc | PNLS |
| OpenAI-o3 (all) | **0.62**    | 0.36        | **0.65**   | 0.59       | **0.90**  | 0.75      | **0.72** | 0.56     |
| GPT-4o (all)    | 0.53        | **0.75**    | 0.57       | **0.65**   | 0.85      | **0.83**  | 0.65     | **0.74** |
| OpenAI-o3       | 0.59        | 0.40        | **0.65**   | 0.61       | 0.84      | 0.78      | 0.69     | 0.60     |
| Gemini-2.5      | 0.46        | 0.63        | 0.60       | 0.64   | 0.77      | 0.79      | 0.61     | 0.69     |
| GPT-4o          | 0.48        | 0.59        | 0.55       | 0.64   | 0.84      | 0.82      | 0.62     | 0.68     |

---

### Author Response · Authors · 2025-12-01
**Final Clarification on Contribution Scope**

As the rebuttal discussion comes to a close, we sincerely thank all reviewers for their helpful feedback and constructive suggestions. The additional experiments and analyses we added—such as o3, Gemini 2.5, human verification, and expanded multilingual statistics—have further strengthened the paper and made its value clearer.

To avoid any remaining misunderstanding, we would like to briefly restate our contribution: VisR-Bench is primarily built on CCPDF, an 8-TB corpus containing about 7.9 million unique PDFs—most of them long, multi-page, high-resolution, and information-dense documents. Unlike conventional large-scale image datasets consisting of single images, each CCPDF document spans multiple info-rich pages containing text blocks, tables, figures, diagrams, and other structured elements. This makes the underlying data fundamentally different and requires substantially more complex processing. Constructing VisR-Bench therefore involved comprehensive multimodal parsing, structured extraction, and QA construction across 16 languages, enabling reliable cross-language and cross-modality evaluation.

In addition, the ColQwen2-E and ColQwen2-M checkpoints we trained provide a controlled analysis of multilingual transfer, showing how additional multilingual supervision influences the learned document-image representations. This comparison also highlights the transfer ability of models trained only on English data when evaluated on multilingual document retrieval.

We hope this clarification helps reviewers and the AC assess our work from the perspective of a large-scale benchmark and dataset contribution.

---

### Meta-Review · Area_Chair_tSeq · 2026-01-05

**Summary:**

The paper proposes VisR-Bench, a benchmark for evaluating visual retrieval-augmented generation in multilingual long documents, comprising over 35K QA pairs derived from a large-scale PDF corpus using the Adobe Document Extract API and GPT-4o for QA generation. Reviewers recognized the benchmark's value in addressing multilingual and multimodal gaps, praising its comprehensive model evaluation and clear presentation. However, major criticisms included the initial omission of reasoning-capable MLLMs like OpenAI o3, overemphasis on Top-1 retrieval metrics, insufficient human validation of synthetic QA pairs, potential biases in dataset construction assumptions, and the exclusion of logographic languages such as Chinese, which limits linguistic diversity. In their rebuttal, the authors enhanced the evaluation by incorporating OpenAI o3 and Gemini 2.5, provided human verification results indicating high QA accuracy, and clarified issues like table QA filtering and corpus constraints for Chinese. While these efforts improved methodological transparency, core concerns remain unaddressed, such as the fundamental reliance on LLM-based heuristics without robust error analysis, the marginal novelty of findings compared to existing benchmarks, and persistent questions about cross-language fairness and generalizability. Given that these weaknesses undermine the benchmark's reliability and contribution significance, the paper may fall short of the high standards required for ICLR acceptance.

**Reviewer Concerns:**

The rebuttal successfully tackled several key issues: the inclusion of evaluations for reasoning-capable MLLMs (OpenAI o3 and Gemini 2.5) resolved criticisms about limited model coverage; human verification on a sample of 50 QA pairs alleviated worries about synthetic data quality, showing high accuracy for figure-based QA; statistical evidence on modality distribution across languages partially addressed dataset balance concerns; and clarifications on the exclusion of Chinese due to corpus constraints and on the intent behind "queries without explicit answers" provided necessary context. However, significant concerns remain outstanding: the benchmark's novelty is still debatable, as findings like struggles with tables and low-resource languages may be perceived as incremental rather than groundbreaking; the human validation was limited to a small subset, leaving broader data reliability questions; the error analysis lacks depth and statistical rigor, offering only qualitative insights; and the reliance on Top-1 accuracy, though justified by the authors for practical scenarios, may not fully counter concerns about its applicability in real-world RAG systems. These unresolved issues, combined with the fundamental reliance on LLM-based heuristics without robust error analysis, undermine the benchmark's contribution.

**Reviewer Scores:**

Reviewer 2yjC​ (original score: 4): The rebuttal addressed concerns about human verification by providing sample validation and clarified heuristic assumptions, likely remains at 4, as some core issues were mitigated, but some reservations about novelty may persist.

Reviewer 1xa4​ (original score: 2): The authors comprehensively responded by adding evaluations of reasoning MLLMs (e.g., o3, Gemini 2.5), justifying Top-1 metrics, and providing human checks, which would substantially improve the score to 4.

Reviewer 3A7H​ (original score: 6): The inclusion of Gemini 2.5 results addressed the main weakness, but the lack of logographic languages remains unresolved; thus, the score might remain at 6.

Reviewer pw8V​ (original score: 6): The rebuttal offered human verification data and error analysis examples, which likely addressed data quality concerns. Thus, the score might remain at 6.

---

### Decision · Program_Chairs · 2026-01-26

Reject